# The association between subfoveal choroidal thickness and refractive error in Taiwanese children: A cross-sectional study

Li-Ying Huang[1©], Lu-Ting Yu[2©], Ning-Yi Hsia[1,3¤*], Yi-Ching Hsieh[1], Hui-Ju Lin[1]

**1** Department of Ophthalmology, Eye Center, China Medical University Hospital, China Medical University, Taichung, Taiwan, **2** School of Medicine, College of Medicine, China Medical University, Taichung, Taiwan, **3** Department of Optometry, Asia University, Taichung, Taiwan

☯ These authors contributed equally to this work.
¤ Current address: China Medical University Hospital, Taichung City, Taiwan.
* deepwhite1111@hotmail.com

## Abstract

### Purpose

The aim was to analyze the association of subfoveal choroidal thickness (SFCT) with age, best-corrected visual acuity, refractive error, and axial length in Taiwan pediatric population.

### Methods

A total of 374 eyes in 187 children were enrolled in this retrospective cross-sectional comparative study, who underwent examinations of best-corrected visual acuity (BCVA), cycloplegic refraction, and axial length (AL). Subfoveal choroidal thickness was assessed utilizing spectral domain enhanced depth imaging optical coherence tomography (EDI-OCT), with measurements taken at the subfovea, defined as the distance from the retinal pigment epithelium to the chorioscleral border.

### Results

The mean age was 5.6 ± 1.9 years (range 2–16 years). The cycloplegic spherical equivalent refractive error was between + 7.25 and - 15.25 diopters (D) and cycloplegic sphere power was between + 8.25 and - 11.5 diopters (D). The mean SFCT was 299.0 ± 69.80 μm. The mean axial length was 22.87 ± 1.29 mm. In univariate analysis, SFCT had significant positive correlations with spherical equivalent (SE) and sphere power ($p < 0.05$) and significant negative correlations with age, cylinder power, and axial length ($p < 0.05$). However, after adjusting in the multivariate regression analysis, spherical equivalent, sphere power and age were not independently associated with SFCT. In multivariate analysis, lower cylinder power and longer axial length have

**Data availability statement:** All relevant data are included within the paper of this IRB approved study (IRB number: CMUH112-REC1-009). However, as this study is based on the analysis of clinical data extracted from patient charts at our institution, the original datasets cannot be shared publicly due to ethical restrictions and the potential for identifying sensitive patient information. In accordance with regulations set by the China Medical University Hospital Institutional Review Board and the national Personal Information Protection Act of Taiwan, the data cannot be distributed outside China Medical University Hospital. Requests for data access or confirmation of institutional policies may be directed to the China Medical University Hospital Institutional Review Board Coordinator via email at cmuh.irb@tool.caaumed.org.tw or by phone at +886-4-22052121 ext. 11925. Additional information is also available on our institutional review board website (https://www.cmuh.cmu.edu.tw/Department/Detail?depid=145).

**Funding:** The author(s) received no specific funding for this work.

**Competing interests:** The authors have declared that no competing interests exist.

significant correlations with thinner SFCT. The relationship between best-corrected visual acuity and SFCT was not significant in both analyses.

## Conclusions

This study showed that mean subfoveal choroidal thickness was $299.0 \pm 69.80$ μm among Taiwanese children. The SFCT was thinner in myopic, longer axial length, and lower cylinder power eyes.

---

## Introduction

The choroid plays a vital role in preserving the health of photoreceptors by providing vascular support and essential nutrients to the outer retina, while also eliminating waste products from this region. Since technology of choroidal thickness measurement by Enhanced Depth Imaging (EDI) mode of optical coherence tomography (OCT) developed in 2008, more detailed and precise choroidal structure can be approached noninvasive [1,2]. Several studies have reported the association between choroidal thickness and visual acuity, age, and myopia [3–11]. Especially in high myopia patients, thinner choroidal thickness was noted with higher myopia.

Myopia is common seen in children, especially among Asians. Studies indicate that increased indoor activities, particularly those involving near-focused tasks like computer use, video gaming, and reading, are associated with elevated incidences of myopia among children [12]. According to the report of National Taiwan University Hospital, there are two peaks of myopia onset, which are 7–8 years old and 13–14 years old. Myopia increases about 100 degrees annually during 7–14 years old, and slowly progression after that. The earlier myopia onset, the faster myopia progression. Myopia is linked with various complications such as chorioretinal atrophy, foveoschisis, choroidal neovascularization, retinal detachment (RD), cataracts, and open-angle glaucoma (OAG). Some of these complications can significantly impair visual acuity, leading to a poor visual prognosis. Currently, four promising treatments for controlling myopia are emerging: Atropine eye drops, Orthokeratology (ortho-k), Myopia control glasses, and Multifocal contact lenses. These treatments aim to modify the structure and focusing of the eye, reducing the strain and fatigue associated with myopia development and progression.

Due to high prevalence of myopia in Asia area, there had been many researches about it in Singapore, China, and South Korea. A study from Singapore, Gupta et al. reported extremely high myopic eyes had thinner choroid [13]. The mean AL was $23.65 \pm 0.89$ mm and the mean age of their study was $21.96 \pm 0.89$ years. A study from Japan that was conducted by Fujiwara et al. reported SFCT depends on age and refractive error [5]. A study from Korea, Lee et al. found choroidal thickness were greater in the hyperopic populations than in the emmetropic and myopic populations [4]. A study from China, Zhu et al. investigated the 52 Chinese children, they reported the thicker SFCT in hyperopia children, and correlated with shorter axial and higher spherical equivalent [9]. Most studies consistently showed that higher myopia is

associated with thinner SFCT. However, these studies covered different age groups and had inconsistent results about choroidal thickness and other ocular characteristics. Studies on children are also rare, with most studies focusing on young adults. Furthermore, there have been limited data of the choroidal thickness in Taiwanese children. Due to above, we aimed to analyze the association of SFCT with factors including age, best-corrected visual acuity, refractive error, and axial length in Taiwan pediatric population; also compared with different countries.

## Methods

### Subjects and enrollment criteria

This was a retrospective cohort study of children's choroidal thickness. The study was conducted in accordance with the Declaration of Helsinki, and approved by the Research Ethics Committee (REC) of the China Medical University and Hospital. Informed consent was not required and was waived by the REC due to the retrospective nature of the study and the use of anonymous data. Medical records of Children under 16-years-old visited Ophthalmic center in China medical university hospital from 01/08/2019–08/31/2022 were included. The data used in this study were accessed between 03/20/2023 and 03/19/2024. Patient who had previously diagnosed retinal pathology, received myopia control treatment or incomplete examination were excluded to prevent interference results and selection bias.

Demographic data of age and sex, and examination data including refractive error after pupil dilation, visual acuity, and axial length were collected. For cycloplegic refraction, 3 drops of 0.5% Tropicamide (Mydriacyl Eye Drops; Alcon, New Zealand) was instillation in each eye at 5-minute intervals. Refractive error was measured after 30 minutes while pupil dilated and loss light reflex. The SE was determined by adding the spherical power to half of the cylinder power. Visual acuity recorded in Snellen were converted into logMAR for analytical purposes. The enhanced depth imaging mode on the spectral domain optical coherence tomography (SD-OCT, Heidelberg Spectralis OCT) was used for subfoveal choroidal thickness measurement. The choroidal thickness was defined as the distance between the retinal pigment epithelium and the inner scleral margin. All measurements were taken by two experienced ophthalmologists simultaneously, who were masked to the other parameters of the patient. Ony the measurements results agreed by both ophthalmologists will be recorded.

We reassign the case study with a research identification number for presentation purposes, without including the case's name or any personally identifiable information. In the subsequent data analysis, the case will be identified solely by the research number, with no personal identifiers such as medical record numbers or names displayed.

### Statistical analysis

Statistical analysis was performed with SPSS software package version 25 (SPSS Inc, Chicago, IL, USA).

Subgroup analysis of SFCT, refractive error and axial length was performed in eyes of different gender and ages using an independent t-tests. Univariate and multivariate regression analysis were used to evaluate the association of potential factors and SFCT. Patient was divided into three groups based on refractive errors. Myopia is defined by SE less than or equal to -0.5 diopters. Hyperopia is defined by SE more than or equal to +0.5 diopters. Emmetropia is defined by SE between -0.5 to +0.5 diopters. One-way analysis of variance with the Bonferroni and Tukey HSD post-test was used to evaluate the difference of SFCT between the three groups. For all the tests, $P < 0.05$ was considered significant.

## Results

### Demographic and baseline characteristics of the subjects

A total of 374 eyes in 187 children were enrolled in this retrospective cohort study. The mean age was $5.6 \pm 1.9$ years (range 2–16 years). The mean cycloplegic spherical equivalent refractive error, mean spherical power and mean cylindrical power were $-0.55 \pm 2.81$ D, $0.23 \pm 2.73$ D, and $-1.44 \pm 1.15$ D, respectively. The cycloplegic spherical equivalent

**Table 1. Patient characteristics.**

|  | Mean±SD | range |
|---|---|---|
| Age, years | 5.64±1.93 | 2–16 |
| Spherical equivalent, D | -0.55±2.81 | -15.25 to 7.25 |
| Spherical power, D | 0.23±2.73 | -11.5 to 8.75 |
| Cylinder power, D | -1.44±1.15 | -6.00 to 1.00 |
| Axial length, mm | 22.87±1.29 | 19.77 to 28.04 |
| BCVA, logMAR | 0.12±0.20 | 0.0 to 1.0 |
| Subfoveal choroidal thickness, um | 299.0±69.80 | 100–500 |

refractive error was between + 7.25 and - 15.25 diopters (D) and cycloplegic sphere power was between + 8.25 and - 11.5 diopters (D). The mean BCVA in LogMAR was 0.12±0.20. The mean subfoveal choroidal thickness (SFCT) was 299.0±69.80 μm. The mean axial length was 22.87±1.29 mm. The demographic characteristics of the patients are summarized in Table 1.

There was no significant association of age, spherical equivalent, cylinder power, visual acuity, and SFCT in genders. Mean spherical power in boys was -0.03±3.00 D, in girls was 0.56±2.3 D. Mean axial length in boys was 23.18±1.33 mm, in girls was 22.46±1.10 mm. There was significant association of spherical power and axial length between two genders, but no significant association of SFCT between genders Table 2 shows the detailed results of the analysis.

There was no significant association of spherical equivalent, spherical power, cylinder power, axial length and SFCT in two age groups. Age more than 5 years old had significant better visual acuity. Table 3 shows the detailed results of the analysis.

## Factors associated with SFCT with univariate and multiple linear regression analysis

Univariate regression analyses showed that age, SE, spherical power, cylindrical power, and AL factors were associated significantly with SFCT (all p<0.05). SFCT had significant positive correlations with spherical equivalent and sphere power and significant negative correlations with age, cylinder power, and AL (all p<0.05). Relationship between the SFCT and demographic factors determined through univariate analyses showed the $R^2$ of age is 0.046, SE is 0.156, AL is 0.198, sphere power is 0.204, and cylinder power is 0.066.

**Table 2. Baseline values and SFCTs by gender.**

|  | Boys | | Girls | | P value |
|---|---|---|---|---|---|
|  | Mean±SD | Range | Mean±SD | Range |  |
| Number of patients (eyes) | 210 | NA | 164 | NA | NA |
| Age, years | 5.7±2.02 | 2–16 | 5.56±1.82 | 2–12 | 0.477 |
| Spherical equivalent, D | -0.78±3.16 | -15.25 to 7.25 | -1.42±1.16 | -6–1 | 0.073 |
| Spherical power, D | -0.03±3.00 | -11.5 to 8 | 0.56±2.3 | -6.5 to 8.75 | 0.041 |
| Cylinder power, D | -1.45±1.13 | -5.5 to 0 | -1.42±1.16 | -6–1 | 0.755 |
| Axial length, mm | 23.18±1.33 | 20.38 to 28.04 | 22.46±1.10 | 19.77 to 25.6 | 0.000 |
| BCVA, logMAR | 0.11±0.16 | 0 to 0.7 | 0.13±0.16 | 0–1 | 0.467 |
| Subfoveal choroidal thickness, um | 296.25±75.99 | 101–500 | 302.46±60.77 | 100–438 | 0.395 |

Abbreviation: NA, not applicable.

**Table 3. Baseline values and SFCTs by age.**

| | Age≦5 years | | Age>5 years | | P value |
|---|---|---|---|---|---|
| | Mean±SD | Range | Mean±SD | Range | |
| Number of patients (eyes) | 208 | NA | 166 | NA | NA |
| Spherical equivalent, D | -0.43±3.07 | -15.25 to 5.25 | -0.69±2.45 | -8 to 7.25 | 0.247 |
| Spherical power, D | 0.45±2.97 | -11.5 to 8.75 | -0.05±2.37 | -6.75 to 8 | 0.192 |
| Cylinder power, D | -1.56±1.12 | -4.5 to 0 | -1.28±1.16 | -6–1 | 0.812 |
| Axial length, mm | 22.52±1.24 | 19.77 to 28.04 | 23.29±1.21 | 20.38 to 25.7 | 0.310 |
| BCVA, logMAR | 0.16±0.17 | 0–1 | 0.06±0.12 | 0 to 0.7 | 0.000 |
| Subfoveal choroidal thickness, um | 308.53±69.56 | 100–500 | 287±68.23 | 112–476 | 0.887 |

Abbreviation: NA, not applicable.

In this univariate regression analyses, age accounted for 4.3%, SE explained 15.4%, spherical power explained 20.2%, cylindrical power explained 6.4%, and AL 19.6% of variation in choroid thickness.

However, after adjusting by multivariate analysis, the spherical equivalent and spherical power were not associated with SFCT (P=0.220; P=0.790). In the multivariate analysis, lower absolute value of cylinder power (β=-0.324, P=.000) and longer axial length (β=-0.208, P=.007) were associated with thinner SFCT. The relationship between best-corrected visual acuity were not significant in both analyses. Table 4 shows the detailed results of the linear regression analysis.

## Subgroup analysis of myopia, emmetropia and hyperopia group

The basic characteristics and SFCT levels were compared between the patients with myopia, emmetropia and hyperopia. Upon comparing the subfoveal choroidal thickness (SFCT) among these three subgroups, it was evident myopic eyes (mean 278.75±67.04 μm) exhibited a significantly thinner choroid compared to both emmetropic (mean 304.30±61.24 μm) and hyperopic (mean 319.49±70.33 μm) eyes. However, SFCT between emmetropic and hyperopic eyes had no significant difference. Table 5 showed the detailed result of the One-way analysis of variance with the Bonferroni and Tukey HSD post-test.

**Table 4. Regression analysis between SFCT with other factors.**

| Univariate analysis | | | |
|---|---|---|---|
| | Beta | Adjusted R2 | P value |
| Age, years | -0.214 | 0.043 | 0.000 |
| Spherical equivalent, D | 0.395 | 0.154 | 0.000 |
| Spherical power, D | 0.452 | 0.202 | 0.000 |
| Cylinder power, D | -0.257 | 0.064 | 0.000 |
| Axial length, mm | -0.445 | 0.196 | 0.000 |
| BCVA, logMAR | -0.009 | -0.003 | 0.855 |
| **Multiivariate analysis** | | | |
| | Beta | | P value |
| Age, years | -0.068 | | 0.206 |
| Spherical equivalent, D | 0.346 | | 0.220 |
| Spherical power, D | -0.075 | | 0.790 |
| Cylinder power, D | -0.324 | | 0.000 |
| Axial length, mm | -0.208 | | 0.007 |
| BCVA, logMAR | -0.074 | | 0.139 |

**Table 5. One-way analysis of variance with the Bonferroni and Tukey HSD post-test.**

| Post hoc | | | Mean difference (I-J) | Std. error | P value | 95% CI | |
|---|---|---|---|---|---|---|---|
| | | | | | | Lower bound | Upper bound |
| Tukey HSD | Myopia | Emmetropia | -25.55 | 10.17 | 0.03 | -49.47 | -1.62 |
| | | Hyperopia | -40.74 | 7.63 | 0.00 | -58.70 | -22.79 |
| | Emmetropia | Myopia | 25.55 | 10.17 | 0.03 | 1.62 | 49.47 |
| | | Hyperopia | -15.19 | 10.33 | 0.31 | -39.50 | 9.11 |
| | Hyperopia | Myopia | 40.74 | 7.63 | 0.00 | 22.79 | 58.70 |
| | | Emmetropia | 15.19 | 10.33 | 0.31 | -9.11 | 39.50 |
| Bonferroni method | Myopia | Emmetropia | -25.55 | 10.17 | 0.04 | -50.00 | -1.10 |
| | | Hyperopia | -40.74 | 7.63 | 0.00 | -59.09 | -22.39 |
| | Emmetropia | Myopia | 25.55 | 10.17 | 0.04 | 1.10 | 50.00 |
| | | Hyperopia | -15.19 | 10.33 | 0.43 | -40.03 | 9.65 |
| | Hyperopia | Myopia | 40.74 | 7.63 | 0.00 | 22.39 | 59.09 |
| | | Emmetropia | 15.19 | 10.33 | 0.43 | -9.65 | 40.03 |

Myopia is defined by SE less than or equal to -0.5 diopters.

Hyperopia is defined by SE more than or equal to +0.5 diopters.

Emmetropia is defined by SE between -0.5 to +0.5 diopters.

## Discussion

This is the first retrospective cohort study, to analyze subfoveal choroidal thickness in Taiwanese children. We found that boys had higher prevalence of myopia and longer axial length. Children younger then 5 years old had worse BCVA compared with older children. Thinner SFCT was found in Taiwanese children with older age, myopia, longer axial length, and lower cylinder power.

A study from China conducted by Wang et al. reported the mean subfoveal CT as 276.21±64.67 µm in control groups and 200.54±69.39 µm in high myopia groups [14]. Their study included 287 participants, mean age was 23.36±7.40 years in control groups and 22.23±6.50 years in high myopia groups. This study had older population than our study population.

A study from Korea, Lee et al. found mean foveal choroidal thickness were 346.86 µm, 301.97 µm, and 267.46 µm in the hyperopia, emmetropia, and myopia groups [4]. The spherical equivalent was +3.39±1.34 diopter, −0.08±0.50 diopter, −2.83±1.17 diopter in the hyperopia, emmetropia, and myopia groups. The axial length was range from 21.01 to 24.69 mm. The age of the participants in their study was quite comparable to that of our study; the mean age was 8.25±1.69 years. However, the number of participants was lower than our study, and our study had wider range of spherical equivalent and axial length. In addition, the association between other characteristics and SFCT in their study was absent.

A study conducted in Japan by Fujiwara et al. found that the SFCT was 265.5±82.4 µm [5]. The research involved 145 participants, with ages ranging from 5 to 88 years and a mean age of 45.7 years. This study encompassed both pediatric and elderly participants, which likely contributed to a significant variation in choroidal thickness.

The study had most similar mean age with our study was the study of Zhu et al. from China [9]. They investigated the 52 children with mean age 7.00±2.00 years in control groups and 6.13±2.24 years in high hyperopia groups. They reported the mean SFCT in 29 emmetropic eyes as 291.27±38.27 µm, and in 23 high hyperopic eyes as 309.22±53.14 µm. In our study, the mean SFCT is 304.30±61.24 µm in emmetropic eyes, and 319.49±70.33 µm in hyperopic eyes. Our participants had younger ages can explain the thicker SFCT in our results. The number of Zhu's study was relatively smaller than our study.

In a study on choroidal thickness and ocular growth in children [15], the subfoveal choroidal thickness is affected by various factors, such as age, ethnicity, gender, axial length, and intraocular pressure. The etiology of choroidal thickness influenced by ocular parameter had been noted in several studies but no consistent result. A biomechanical stretching by axial elongation and eyeball extension may have a direct effect on the choroidal, retinal, and scleral thinning, especially in high myopic eyes [14]. However, the subfoveal thickness of myopic children was thinner than expected thickness after mechanical stretch by axial elongation. Decreased choroidal blood flow and/or blood vessel area also contribute to the thinner SFCT in myopic children [16,17].

The difference of ocular biometrics between boys and girls was controversial between studies. Our study demonstrated that male had more myopia and longer axial length. Although there was no significant difference of SFCT between genders, male had thinner mean SFCT, which is corresponded to higher myopia and longer axial length in males. Some studies reported that female had thinner SFCT, but others reported no significant differences between boys and girls for SFCT and SER [3,6,10]. Cevher et al. reported females have higher basal sympathetic tone than males which can cause vasoconstriction and induced thinner choroidal thickness [10]. In our studies, mean age is 5-years-old, which is too young to have significant higher sympathetic tone. This may explain the difference between our study and other studies.

Several studies also reported SFCT in younger age was significant thicker than older subjects. Fujiwara and Shiragami found SFCT was thickest in subjects younger than 10 years, and gradually decreased after 30 years old [5]. Study showed the significant negative relationship between age and choroidal thickness, but age between 20–40 years was not affected [10]. Xiong et al reported only children younger than 11 years old had significant negative correlation between age and SFCT. In our study, there is significant negative relationship of age and SFCT in univariate regression analyses, but no significant in multiple linear regression analyses. Based on the above studies, the SFCT had grossly negative relationship with age, but have relative stable thickness during third decades.

Previous studies reported a SFCT of 291–359 µm of children in different races. Zhu et al reported the SFCT of normal Chinese children was 291.27±38.27 µm. [9] Read et al reported the SFCT of normal children in Australia was 359±77 µm [16]. Lee et al reported the SFCT of normal Korean children was 301.97±55.93 µm. [4] In our study, the SFCT of normal Taiwanese children was 304.30±61.24 µm, which was like other Asia children, but thinner than Australia children. The difference might due to the Caucasian children in Australia, which had thicker SFCT. Youngseok Song et al. noted Malays exhibited thinner SFCT compared with Indians and Chinese [18]. Rhodes et al also reported choroidal thickness varies with race, that African descent have a thicker peripapillary choroid than European descent [19]. Although there is no date compare the choroidal thickness between Asian and Europeans, the SFCT between Asians will be more close than non-Asian ones.

Ocular biometrics, such as refractive error, visual acuity, and axial length, as well as demographic factors, including ethnicity, sex, and age, have been found to be associated with SFCT in several studies [3–11]. We studied the relationship between gender and other factors, including spherical equivalent, spherical power, cylinder power, axial length and SFCT. As Fujiwara and Shiragami reported, there was no significant difference in SFCT between the men and women [5].

There is no significant relationship with visual acuity and SFCT in our study. Gupta et al. reported that choroidal thickness was not an independent predictor of visual acuity [13]. SE is more important predictor of BCVA.

Our results showed that the SFCT was significantly associated with age, refractive error, and axial length. Result of refractive error had some disparity between previous studies. To establish whether the associations among the SFCT, demographic profile, and various ocular factors, we conducted both univariate and multivariate linear regression analyses. Significant difference of spherical equivalent, spherical power, cylindrical power, axial length and SFCT was found using univariate linear regression analysis. Spherical power and axial length had highest association of SFCT, which is like previous studies. After adjusting by multivariate linear regression analysis, the cylindrical power and axial length are still significantly associated with SFCT. Therefore, we concluded that the thinner SFCT in higher spherical equivalent and higher spherical power eyes might be secondary in its effects to the longer AL, but they are not independent factors. However,

only cylindrical power and axial length had significant difference of SFCT in multivariate linear regression analysis. The higher absolute value of cylindrical power, the thicker of SCFT. The association of cylinder and SFCT had not been mentioned in previous reports. The clinical practice of using atropine and optical interventions to control myopia and axial length elongation have become more widespread. Most mechanisms of myopia control interventions were associated with higher high-order aberrations (HOAs) and altered retinal image. Higher HOA influenced retinal image quality, which further impact the eye growth and refractive error. Children with higher astigmatism had been reported to have more HOAs [20,21]. Higher corneal HOAs exhibited significant correlations with reduced myopia progression and axial elongation [22]. Based on above, higher cylinder power is associated with higher HOA, which reduced axial elongation and myopia progression, might have thicker SFCT. The result still needs to be confirmed by more longitudinal studies.

Multiple studies have shown that the choroidal thickness in highly myopic eyes is thinner compared to that in normal eyes [15]. Patient was divided into three groups according to spherical equivalent. The SFCT of myopic children was reported to be 278.75 μm in our study, the SFCT in the emmetropic children was 304.30 μm, and the SFCT in the hyperopic children was 319.49 μm. There is significant difference between myopic and other two groups, but no significant difference between emmetropic and hyperopic children.

Our study had several strengths. First, this is the first Taiwanese children population study exploring the association of choroidal thickness. Second, the results of the association of cylindrical power and SFCT was first reported. Third, we had excluded children with ocular disease or received atropine treatment to less selection bias.

There are several limitations in this study. First, the manually measurement method was performed for choroidal thickness in this study, which is not as objective as automated software measurement. But measurements taken by two experienced ophthalmologists can reduce error. Second, the followed series was not acquired to evaluate the consequence change of choroidal thickness during different groups. Therefore, we cannot explicitly link SFCT to predict myopia progression. Further studies with more objective measurement and longitudinal studies are needed for choroidal thickness change. Third, this hospital-based cohort might have selection bias due to patient might have some ocular disease or use medication treatments. Therefore, we excluded patient who had previously diagnosed retinal pathology, received myopia control treatment. However, the hospitals in Taiwan are highly accessible and most Taiwanese children are likely to receive ocular examination and visual screening since preschool age due to the national policy.

## Conclusion

Our findings provide the profiles of SFCT in the eyes of Taiwanese children populations and to offer insights into the early interventions and monitors for thinner SFCT and lower astigmatism. Our findings revealed that thinner SFCT was correlated with older age, myopia, longer axial length, and lower cylinder power. This is the first study reported the association between cylinder power and SFCT. Importantly, SFCT demonstrated potential as a predictor of myopia progression, with tendencies to decrease with longer axial length and lower cylinder power. However, further studies are warranted to evaluate the predictive value of SFCT in myopia progression more comprehensively. These insights underscore the importance of considering SFCT in the assessment and management of myopia in pediatric populations, paving the way for targeted interventions aimed at mitigating myopia progression in children.

## Author contributions

**Conceptualization:** Li-Ying Huang, Ning-Yi Hsia, Yi-Ching Hsieh, Hui-Ju Lin.

**Data curation:** Li-Ying Huang, Lu-Ting Yu, Ning-Yi Hsia, Yi-Ching Hsieh, Hui-Ju Lin.

**Formal analysis:** Lu-Ting Yu, Ning-Yi Hsia.

**Methodology:** Li-Ying Huang.

**Project administration:** Li-Ying Huang.

**Resources:** Ning-Yi Hsia.

**Supervision:** Ning-Yi Hsia.

**Writing – original draft:** Li-Ying Huang, Lu-Ting Yu.

**Writing – review & editing:** Ning-Yi Hsia, Hui-Ju Lin.

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
