## [Decision Letter · Decision Letter 0]

26 Jan 2025

PONE-D-24-56020The Association of Choroidal Thickness and Refractive Error in Taiwanese Children : A cross-sectional study and systemic reviewPLOS ONE

Dear Dr. Hsia,

Thank you for submitting your manuscript to PLOS ONE. After careful consideration, we feel that it has merit but does not fully meet PLOS ONE’s publication criteria as it currently stands. Therefore, we invite you to submit a revised version of the manuscript that addresses the points raised during the review process. Please submit your revised manuscript by Mar 12 2025 11:59PM. If you will need more time than this to complete your revisions, please reply to this message or contact the journal office at plosone@plos.org . Please include the following items when submitting your revised manuscript:

We look forward to receiving your revised manuscript.

Kind regards,

Jiro Kogo

Academic Editor

PLOS ONE

Journal Requirements:

2. During your revisions, please note that a simple title correction is required: “The Association of Choroidal Thickness and Refractive Error in Taiwanese Children : A cross-sectional study’. Please ensure this is updated in the manuscript file and the online submission information.

3. You indicated that you had ethical approval for your study. In your Methods section, please ensure you have also stated whether you obtained consent from parents or guardians of the minors included in the study or whether the research ethics committee or IRB specifically waived the need for their consent.

Reviewers' comments:

Reviewer's Responses to Questions

**Comments to the Author**

1. Is the manuscript technically sound, and do the data support the conclusions?

Reviewer #1: Yes

Reviewer #2: Yes

2. Has the statistical analysis been performed appropriately and rigorously? 

Reviewer #1: Yes

Reviewer #2: Yes

3. Have the authors made all data underlying the findings in their manuscript fully available?

Reviewer #1: Yes

Reviewer #2: No

4. Is the manuscript presented in an intelligible fashion and written in standard English?

Reviewer #1: Yes

Reviewer #2: Yes

5. Review Comments to the Author

Reviewer #1: The manuscript explores the association of subfoveal choroidal thickness (SFCT) with various ocular and demographic factors in Taiwanese children, an important area considering the high prevalence of myopia in Asia. While the study adds value, particularly in a pediatric Taiwanese cohort, several areas require clarification or improvement.

Title

1. Title Clarity: Consider revising the title to "The Association Between Subfoveal Choroidal Thickness and Refractive Error in Taiwanese Children: A Cross-sectional Study and Systematic Review." This emphasizes the study's primary focus and methods.

Abstract

1. Methods: The abstract should explicitly include the study's design (e.g., "retrospective cross-sectional study").

2. Results: The key findings are clear but could emphasize multivariate analysis outcomes, as they are more robust.

3. Conclusion: Avoid speculative phrases like "might be a good predictor." Instead, state the significance of findings confidently while reserving speculation for the discussion.

Introduction

1. Literature Context: Expand on how the study differs from previous works in Chinese, Korean, or Australian children to establish the novelty better.

2. Aim Clarity: The aim is well stated, but the introduction would benefit from explicitly linking SFCT to its clinical implications in predicting myopia progression.

Methods

1. Study Design: Clarify why a retrospective design was chosen and discuss the potential limitations of retrospective data.

2. Definitions: Provide more details about cycloplegic refraction procedures (e.g., drugs used, cycloplegic protocol).

3. Exclusions: Add rationale for excluding patients with incomplete exams or prior myopia control treatments.

4. Measurement: Mention the interobserver variability or agreement for manual SFCT measurements.

Results

1. Demographics: The mean age and axial length provide useful context, but gender-based differences in axial length (significant) and SFCT (non-significant) should be discussed more comprehensively.

2. Regression Analysis: The multivariate analysis results should be emphasized more, as they account for confounders. Consider rephrasing for clarity.

Discussion

1. Comparison with Literature: The study compares findings with data from other regions but needs to delve deeper into why SFCT in Taiwanese children aligns more closely with other Asian cohorts than non-Asian ones.

2. Clinical Implications: Discuss how these findings could influence myopia management in children (e.g., early interventions for thinner SFCT).

3. Cylindrical Power: The association of SFCT with cylindrical power is novel but requires more discussion. Could this be a surrogate for corneal or lenticular changes in myopic eyes?

4. Limitations: The discussion acknowledges the lack of longitudinal data and manual SFCT measurements but should also address potential selection bias in a hospital-based cohort.

Reviewer #2: Comments to the Editor of PLOS ONE

Dear Editor,

Thank you for the opportunity to review this manuscript. Please find my comments below.

Introduction

The manuscript does not provide a compelling justification for the need of the study. It would benefit from a more robust rationale for why the research is important. Additionally, the background section should better explain the potential relationship between choroidal thickness, myopia, and other relevant parameters measured in the study.

Methods

The criteria for including and excluding participants in the study were not sufficiently specified. It is important to provide clearer and more detailed information on the selection process. Furthermore, a more thorough description of how the clinical assessments were conducted is necessary to ensure transparency and reproducibility.

Discussion

The discussion section is lacking a comprehensive analysis of the study's findings in comparison with previous research. It would be valuable to include a comparison with existing literature, noting any similarities or discrepancies, and providing potential explanations for these observations. Furthermore, the discussion should be revised to avoid redundancy and repetition of the results section.

Strengths and Limitations

While the manuscript addresses some limitations of the study, it does not mention its strengths. It would be helpful to provide a more balanced view by discussing the study’s strengths in addition to its limitations.

Recommendation

Overall, the study presents some merits. However, I recommend a major revision to address the points outlined above

6. PLOS authors have the option to publish the peer review history of their article (what does this mean? ). If published, this will include your full peer review and any attached files.

**Do you want your identity to be public for this peer review?** For information about this choice, including consent withdrawal, please see our Privacy Policy .

Reviewer #1: No

Reviewer #2: **Yes: ** Ngozika Esther Ezinne

---

## [Author Response · Author response to Decision Letter 1]

7 Mar 2025

Thank you for your comments. We have gone through your comments carefully and tried our best to address them one by one. We hope the manuscript has been improved accordingly.

---

## [Decision Letter · Decision Letter 1]

31 Mar 2025

The Association Between Subfoveal Choroidal Thickness and Refractive Error in Taiwanese Children : A cross-sectional study

PONE-D-24-56020R1

Dear Dr. Hsia

We’re pleased to inform you that your manuscript has been judged scientifically suitable for publication and will be formally accepted for publication once it meets all outstanding technical requirements.

Kind regards,

Jiro Kogo

Academic Editor

PLOS ONE

Additional Editor Comments (optional):

Reviewers' comments:

Reviewer's Responses to Questions

**Comments to the Author**

1. If the authors have adequately addressed your comments raised in a previous round of review and you feel that this manuscript is now acceptable for publication, you may indicate that here to bypass the “Comments to the Author” section, enter your conflict of interest statement in the “Confidential to Editor” section, and submit your "Accept" recommendation.

Reviewer #2: All comments have been addressed

2. Is the manuscript technically sound, and do the data support the conclusions?

Reviewer #2: Yes

3. Has the statistical analysis been performed appropriately and rigorously? 

Reviewer #2: Yes

4. Have the authors made all data underlying the findings in their manuscript fully available?

Reviewer #2: No

5. Is the manuscript presented in an intelligible fashion and written in standard English?

Reviewer #2: Yes

6. Review Comments to the Author

Reviewer #2: The authors have done commendable work by addressing all the comments, which has significantly improved the quality of the manuscript. However, I recommend a comprehensive review of the English language to further enhance the overall quality of the write-up.

7. PLOS authors have the option to publish the peer review history of their article (what does this mean? ). If published, this will include your full peer review and any attached files.

**Do you want your identity to be public for this peer review?** For information about this choice, including consent withdrawal, please see our Privacy Policy .

Reviewer #2: **Yes: ** Ngozika Esther Ezinne

---

## [Editor Report · Acceptance letter]

PONE-D-24-56020R1

PLOS ONE

Dear Dr. Hsia,

I'm pleased to inform you that your manuscript has been deemed suitable for publication in PLOS ONE. Congratulations! Your manuscript is now being handed over to our production team.

Kind regards,

on behalf of

Prof. Jiro Kogo

Academic Editor

PLOS ONE